

# Emerging Mineral Dust Source in 'A'ặy Chù' Valley, Yukon, Canada Poses Potential Health Risk via Exposure to Metal and Metalloids Enriched in PM$_{10}$ and PM$_{2.5}$ Size Fractions

Arnold R. Downey[1], Alisée Dourlent[1,2], Daniel Bellamy[3], James King[3], Patrick L. Hayes[1]

[1]Department of Chemistry, Faculty of Arts and Sciences, Université de Montréal, Montréal, H2V0B3, Canada.
[2]Departement de la chimie, Faculté des Sciences et Ingénierie, Université de Sorbonne, Sorbonne, 75005, France
[3]Department of Geography, Faculty of Arts and Sciences, Université de Montréal, Montréal, H2V0B3, Canada

*Correspondence to:* Patrick L. Hayes (patrick.hayes@umontreal.ca)



**Abstract.** The 'A'äy Chù' Valley in Kluane National Park and Reserve, Yukon, Canada has undergone significant hydrological change in the past decade due to climate-driven glacial recession. This has reverted the 'A'äy Chù' to a major source of sediment-derived mineral dust, representing an environmental change for the region. Mineral dust influences climatic radiative forcing and impacts human health, both of which depend on its concentration, size distribution, and composition. This work discusses results from a field campaign conducted in the 'A'äy Chù' Valley in 2021 aimed at understanding and quantifying these parameters, with comparison to a previous campaign in the same location to evaluate the evolution of the dust emissions between 2018 and 2021. An optical particle counter (OPC) instrument measured a mean volume diameter of airborne dust of 4.43 μm at 3.3 m above ground, with Coulter Counter measurements being used for comparison and validation. The concentration of many metal(loid)s in the dust were also studied: Al, Ag, As, Ba, Ca, Cd, Co, Cr, Cu, Fe, K, Mg, Mn, Ni, Pb, Rb, Tl, U, V, and Zn. It was found that 24-hour ambient air quality criteria for exposure to several metal(loid)s were surpassed. Significant enrichment of several metal(loid)s was observed for both the $PM_{10}$ and $PM_{2.5}$ size fractions relative to the Total Suspended Material (TSP) fraction of the mineral dust. This suggests that the mineral dust in the 'A'äy Chù' Valley possesses compounding characteristics that are detrimental to human health due to exposure to potentially toxic metal(loid) concentrations.

**Keywords:** High-Latitude Mineral Dust, Elemental Analysis, Size Distribution, Diurnal Variation, Meteorology, Glacial Sedimentology, 'A'äy Chù' Valley, Atmospheric Aerosols, Air Quality, Climate Change

## 1. Introduction

*The field campaign for this project was conducted on the traditional territories of the Champagne and Aishihik First Nations and Kluane First Nation. These peoples are included in the Southern Tutchone First Nations. Therefore, the Southern Tutchone names and spellings of places referenced in this work are used.*

The size distribution of aerosols, including mineral dust, is one of the key properties determining their impact on climate and health. The radiative properties of aerosol particles are intimately tied to size, with scattering cross section scaling as a power of 3-5 with respect to particle diameter (Knippertz and Stuut, 2014; Tegen and Lacis, 1996). For mineral dust, particles with sizes on the order of solar short-wave radiation wavelengths (0.2-2 μm) produce the greatest shortwave radiative effect per unit mass and generally have a direct cooling effect (Miller et al., 2006). Similarly, mineral dust particles with sizes on the order of terrestrial radiation wavelengths, >4 μm, produce the greatest warming effect (Tegen and Lacis, 1996). The indirect radiative properties are also affected by size, such as when aerosol particles act as Cloud Condensation Nuclei (CCN) or Ice Nulei (IN) (Xi et al., 2022; Kok et al., 2023). An aerosol's role in multiphase chemical reactions is also affected by size, where the particle's surface area and surface-to-volume ratio are key parameters (Fang, 2018). All of these effects are related to atmospheric lifetime, which is also primarily driven by aerosol particle size. Atmospheric lifetimes of dust particles have been



modelled based on effective radius ($r_{eff}$), accounting for both wet and dry deposition processes. For instance, large sand-sized particles with $r_{eff} = 38$ μm give a modeled lifetime of only 1 hour, while small clay-sized particles with $r_{eff} = 0.7$ μm give a modeled lifetime of 13 days (Tegen and Fung, 1994).

70         Furthermore, the impact of particulate matter (PM) on health is related to particle size. It is known that coarse particles larger than 10 μm typically deposit in the oral and nasal cavities upon respiration. Particles less than 10 μm in diameter travel further into the respiratory system, exhibiting greater deposition traction in the bronchial and alveolar regions of the lungs, with this effect being further pronounced for particles of less than 2.5 μm (PM$_{2.5}$) (Hofmann, 2011). A systematic review on PM and all non-accidental mortality reported a higher risk on a per-mass basis associated with PM$_{2.5}$ relative to particles less than 10 μm in diameter (PM$_{10}$), due to lung cancer, Chronic Obstructive Pulmonary Disease and other negative

health outcomes (Chen and Hoek, 2020; World Health Organization, 2021). Thus, the daily exposure limit recommended by the WHO for PM$_{2.5}$ is much lower than that for PM$_{10}$, 15 versus 45 μg m$^{-3}$, respectively (World Health Organization, 2021). One should note that PM$_{2.5}$ naturally contains more particles per volume, with greater specific surface area than PM$_{10}$. It is also known that the composition of PM plays a role in its impact on human health. Chen et al. outlines this phenomenon, where the hazard risk associated with certain components of PM is quite high (Ni, V, EC) compared to others (Si, Al, NO$_3$)

(Chen and Lippmann, 2009). Environmental and Public Health organizations thus establish exposure limits to potentially toxic PM components like metals and metalloids typically based on epidemiological studies, with each component possessing its own toxicological characteristics (Ali et al., 2019).

        For mineral dust, size distributions span multiple orders of magnitude, with diameters ranging from less than 100 nm to more than 100 μm (Marticorena, 2014), making complete assessments of the particle size distribution difficult. Nonetheless,

a great deal of research has been conducted to assess dust size distributions of geographically diverse sources (Scheuvens and Kandler, 2014). Mid-latitude dust sources have received the most attention since these sources contribute to the vast majority of earth's yearly dust budget. High-latitude dust (HLD) sources are estimated to only contribute 1-5% of global dust emissions as compiled by Meinander et al (Meinander et al., 2022; Dagsson-Waldhauserova et al., 2019). However, these dust sources demonstrate climatic and environmental significance and should not be overlooked. Although HLD contributes a small

percentage to the Earth's global dust budget, it is estimated to account for 57% of the dust deposited on snow and ice surfaces (Meinander et al., 2022). This causes the albedo of these surfaces to decrease substantially, suggesting that HLD may have a disproportionate impact on climate (Boy et al., 2019). The dust-induced snow albedo effect is influenced by many factors, including its concentration in the snow and dust optical properties, as determined by its size distribution and chemical



composition (Flanner et al., 2009; Dang et al., 2015; Flanner et al., 2021). Mineral dust with greater concentrations of iron

oxide species, for example, is known to have a greater imaginary refractive index, leading to a higher tendency to absorb

radiation (Zhang et al., 2015a), causing warming and accelerated melting of ice and snow. The composition and size

characteristics of HLD have been studied in recent years using a wide variety of techniques, as summarized  in Table 1 of

Meinander et al. 2022 (Meinander et al., 2022).

HLD composition has been characterized to some extent on a size-resolved basis with respect to mineral content

(Kandler et al., 2020; Barr et al., 2023). However, the direct relationship between the metallic composition of mineral dusts

and their size class in post-glaciated high-latitude regions has not yet been characterized, to our knowledge. Some insight into

the composition of HLD in different size fractions can be derived from previous research related to drift prospecting for

mining purposes (Shilts, 1993). Metal and metalloid concentrations in glacial sediments increase substantially in sediments

of finer size fractions due to the tendency of minerals to crush to specific sizes during comminution, which is the reduction

of geological material to smaller average sizes during glacial transport. We refer to this increase in metal(loid) concentration

as "enrichment". Enrichment has been noted for several metal(loid)s studied, reaching maximal concentrations in size

fractions below 4 μm in diameter, but greater than 1 μm *(Shilts, 1984a)*. The metal(loid)s are thought to reside within the

lattices of the physically comminuted phyllosilicates that are abundant in these size fractions and the observed enrichment

occurs in both weathered and unweathered sediment *(Shilts, 1984b)*.

Thus, a complete understanding of the size distribution and metallic composition of emerging HLD sources is

desirable from climate, geo-chemical, and public health standpoints. As noted, mineral dust size distributions are notoriously

difficult to measure, owing to their large size distribution range, along with various shortcomings of aerosol size measurement

techniques (Reid et al., 2003). Aerosol light scattering is often exploited to measure aerosol size distributions in real time,

such as with an Optical Particle Counter (OPC) (Mcmurry, 2000). Such instruments are efficient at measuring particle size

distributions at high temporal resolution and are relatively low-maintenance and robust for a variety of field applications. The

relationship between particle size and light scattering is mathematically complicated, yet direct for spherical particles of

known composition thanks to applications of Mie scattering theory (Wriedt, 2012). This is not the case, however, for mineral

dust. Mineral dust particles adopt a variety of shapes, and are composed of several different minerals, meaning their interaction

with the light sources of OPC instruments is not straightforward (Scheuvens and Kandler, 2014). The varied composition of

mineral dust can furthermore lead to slight oversizing of particles, while the effect of shape depends on scattering angle

(Collins et al., 2000). Some degree of confidence on size distribution results can be obtained by comparison with other sizing




techniques, such as using a Coulter Counter instrument (Davies, 1970). Coulter Counter instruments measure particle size distributions by resistive pulse sensing as particles pass through a sensing region while suspended in an electrolyte solution. The magnitude of the resistive pulse it proportional to the volume of the particle in the sensing region from which a high-resolution particle size distribution may be obtained containing hundreds of size bins (Mctainsh et al., 1997). Although the resistive pulse is proportional to the particle volume, this technique is still susceptible to biases from non-sphericity (Hurley, 1970). Another drawback to such a technique is that it must be performed off-line, requiring sampling and sample processing steps. This means that high time resolution is not obtainable, as samples are usually collected over the course of one or two days. It also means that the particles are not in their natural chemical environment during analysis, with some components possibly dissolving or disaggregating in the electrolyte solution. It is thus an advantage to assess mineral dust size distributions using parallel techniques, when possible, to compensate for each technique's shortcomings.

The mechanisms dictating the production of mineral dust are relatively well-established. Aeolian erosion is related to wind shear stress, which is related to the gradient of wind speed with height and the dynamical viscosity of the air.(Marticorena, 2014) A wind speed threshold must be met to overcome the forces holding particles in place, namely their weight, and interparticle cohesion forces resulting from electrostatics and soil moisture (Shao and Lu, 2000; Kok et al., 2012). For example, a typical threshold wind speed used in dust emission modelling is 6.5 m s$^{-1}$ at 10 m height, mainly due to lack of input on surface properties at the global scale. Below this threshold wind speed, one does not expect any fine dust emission (Sokolik, 2002; Kalma et al., 1988). In comparison, the erosion thresholds for wind velocities at 10 m measured on natural surfaces range from 4 to 20 m s$^{-1}$ (Nickling and Gillies, 1989; Helgren and Prospero, 1987). To produce dust, these winds must act on a bare surface of soil or sediment that is relatively dry. As such, dust emission models commonly operate as a function of surface conditions and shear velocity (Lee et al., 2019). A better understanding of the meteorological conditions of the 'A'ąy Chù' Valley in relation to dust emissions would provide insights leading to better integration of these types of emissions into climate and geochemical transport models. The present study provides unique observations that can be used for such model improvements.

Overall, it has been determined that climate change likely enhances dust activity to a greater extent than it reduces it. Dust emissions are estimated to have increased by $55 \pm 30\%$ since pre-industrial era (Kok et al., 2023), and dust concentrations are expected to increase 10% more by 2100 according to the IPCC (Zhai et al., 2021). Hydrological changes driven by climate change are one mechanism by which dust activity may increase in the future. The river piracy event experienced by the 'A'ąy Chù' (formerly Slims River) in 2016 is a dramatic example of such a change.  Located in Kluane





National Park and Reserve in southwest Yukon, Canada, the 'A'äy Chù' Valley runs from the foot of the Kaskawulsh Glacier

to Lhù'ààn Mân' (Kluane Lake). At 409 km², Lhù'ààn Mân' is the largest lake contained entirely in the Yukon (Natural

Resources Canada et al., 1973). The valley is over 30 km long from the tip of the Kaskawulsh to the edge of the delta before

the lake. The recession of the Kaskawulsh Glacier has been documented since the 1970s (Foy et al., 2011; Denton and Stuiver,

1966). In 2016, this recession reached a point where the majority of the 'A'äy Chù' river flow was diverted to the Kaskawulsh

River(Shugar et al., 2017) leaving the 'A'äy Chù' as a dried sediment bed. The sediment there is exposed to strong winds in

the valley, and as a result, dust activity has increased substantially in the southern Lhù'ààn Mân' region (Bachelder et al.,

2020).

The goal of this study is to accurately describe the size distribution, diurnal trend, and size-resolved composition of

mineral dust emissions from the 'A'äy Chù' Valley. This emerging dust source has been a recent topic of study for both field-

based and remote sensing-based research. Remote sensing approaches involving LIDAR have been useful for characterizing

large-scale vertical extent and optical properties of the dust (Sayedain et al., 2023). A previous field-based measurement

campaign was successful in providing a preliminary characterization of the composition and size distribution of the dust from

this source and noted several elemental enrichments relative to the source soil (Bachelder et al., 2020). This work did not

directly study the finer $PM_{2.5}$ size class composition or the non-size-discriminated Total Suspended Material (TSP) fraction

and was relatively limited in terms of the elements that were analyzed. For instance, this study did not quantify some elements

of importance for human health considerations, like Ba, Cr, and V. As mineral dust transport is often implicated in nutrient

transport modelling, it is also important to quantify components like Ca and Mg (Schepanski, 2018; Zhang et al., 2015b).

Furthermore, the size distributions measured by OPC were not compared and validated against another technique. Therefore,

the present study aims to expand on the progress that has been made on understanding this unique mineral dust source and to

provide a basis for future characterization of high-latitude mineral dust.

**2. Methods**

**2.1 Field Site and Sampling Approach.** A map of the region where sampling was performed is presented in Fig. 1. The GPS

coordinates of the sampling location, hereafter referred to as the Down Valley Site, are 60°59'55.25"N, 138°31'24.05"W. This

sampling location has been previously used in a number of recent studies also studying mineral dust dynamics and properties

(Bachelder et al., 2020; Sayedain et al., 2023; Bellamy et al., 2025a; Bellamy et al., 2025b; Tardif et al., 2025). PM sampling

began on June 4th and ended on July 2nd, 2021. It was noted from previous campaigns in this location that late spring and

early summer show the highest dust activity, due to a combination of the ground being thawed following the winter season



and the melt water from the glacier and snow of adjacent mountains not yet being so abundant as to moisten the sediment (Bachelder et al., 2020).


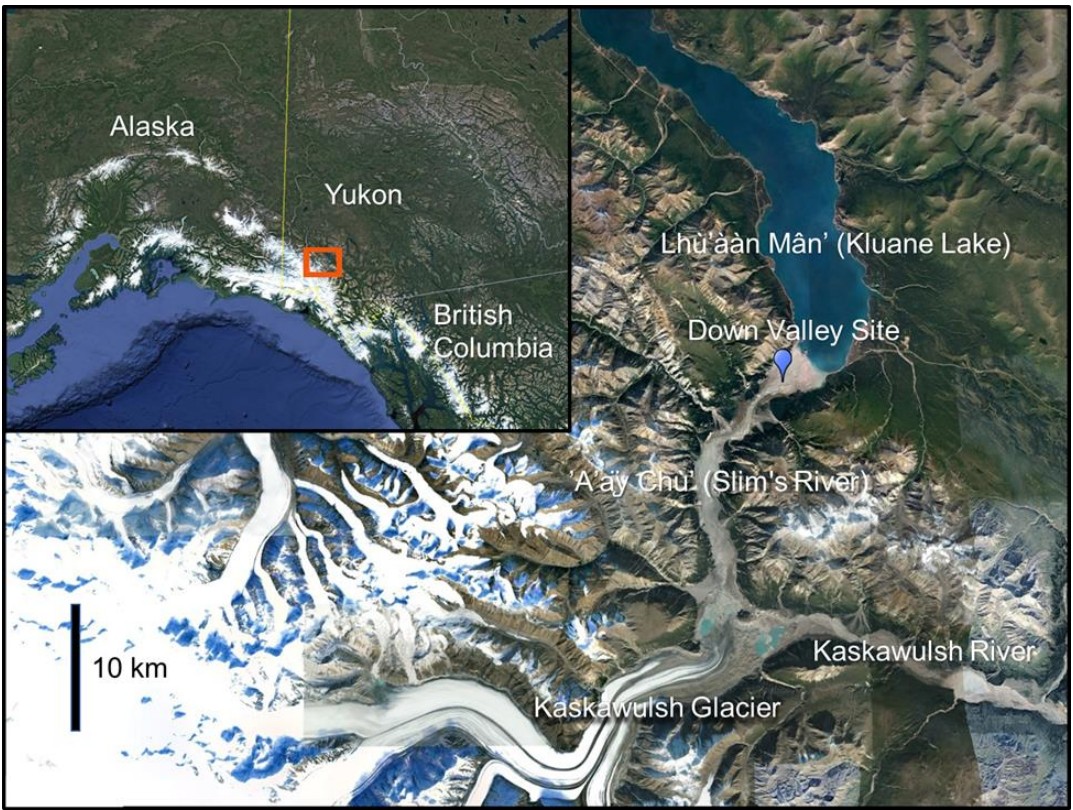

**Figure 1. Map of sampling location. Samples were collected at the Down Valley Site. The insert map shows the location of the area in Yukon, Canada. Maps produced by ©2023 Google Earth.**

PM sampling was conducted using a set of four ARA N-FRM Samplers (ARA Instruments) positioned in pairs at

3.3 m and 6.3 m above the ground, with respect to the height of the sampling head inlets, Fig. 2. The sampling was conducted at different heights to provide information on the vertical gradient of PM concentrations. The samplers were set in pairs to allow for simultaneous sampling at a given height of different PM size classes, Total Suspended Particles (TSP), $PM_{10}$, and $PM_{2.5}$. PM sampling was conducted on nearly a daily basis for TSP and $PM_{10}$ samples, using an approximate collection time of 24 hours, whereas a 48-hour collection time was used for $PM_{2.5}$ samples to obtain greater masses of $PM_{2.5}$ for improved

quantitation of trace elements. A flow rate of 16.7 L $min^{-1}$ was used for samples destined for metal(loid)s analysis. All PM



sampled this way was either collected on Teflon (PALL Corporation, Teflo, 2.0 μm pore size, 47 mm diameter) or Quartz fiber (Whatman™, Grade QMA, 47 mm diameter) filters. Filters were weighed and pre-weighed using a microbalance (VWR, VWR-21XC, readability 1 μg). Quality control procedures for gravimetric measurements were adapted from the US-EPA's Quality Assurance Guidance Document 2.12 for Monitoring $PM_{2.5}$ in Ambient Air Using Designated Reference or Class 1

Equivalent Methods (U.S. Epa, 2016). As such, at least one lab blank and one field blank filter was reserved for every 10 filters used for sampling. The filters were conditioned in the weighing room for at least 24 hours prior to their weighing or pre-weighing. The temperature and relative humidity (RH) of the weighing room was monitored during the conditioning and weighing periods. The specifications on average temperature (between 20 and 23 °C), temperature variability (±2 °C), and RH variability (±5%) for the 24 hours prior to weighing sessions were always met, while the specification for average RH

(between 30 and 40%) was sometimes not met, with RH exceeding 40% upon re-weighing. Following gravimetric measurements, the filter samples were stored at 4 °C prior to preparation for elemental analysis. Any value deemed to be below the Limit of Detection (LOD) was assumed to be one half of the LOD for the given size class, determined using the collected field blanks.

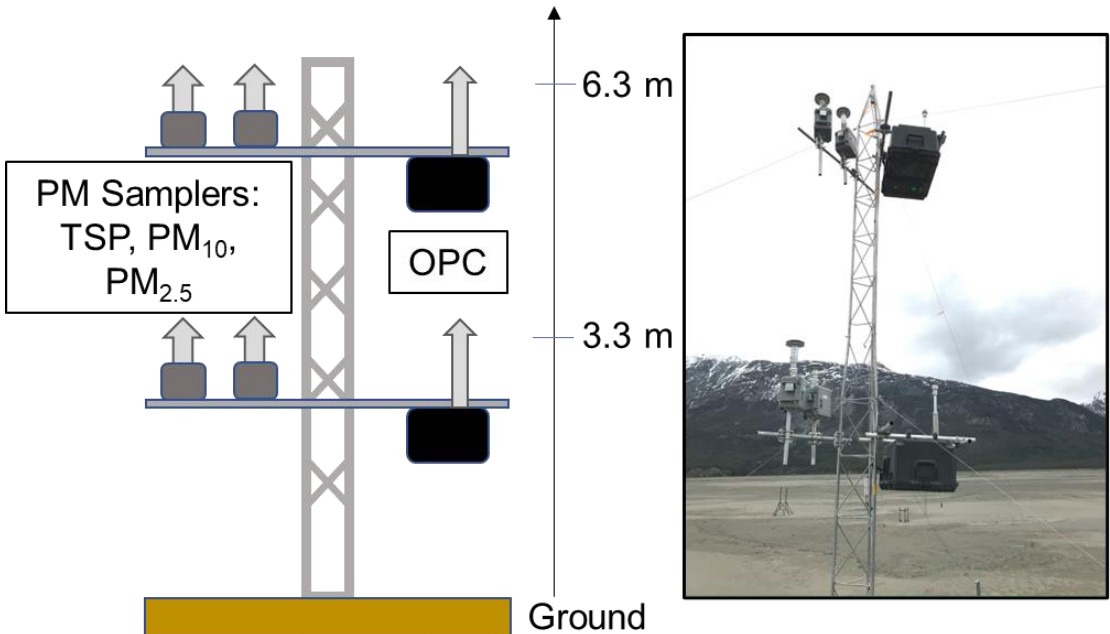

**Figure 2. Schematic and photo of the dust sampling tower. Measurements were conducted from June 4ᵗʰ to July 2ⁿᵈ, 2021 in 'A'ąy Chù' Valley, Kluane National Park and Reserve, Yukon.**



**2.2 Sample Preparation.** Filter samples were prepared for total metal(loid)s determination following a modified version of US EPA method 3051a: Microwave Assisted Acid Digestion of Sediments, Sludges, and Oils (U.S. Epa, 2007). One half of a sample filter was transferred to a previously washed PFA microwave digestion vessel. Next, 4 mL of concentrated nitric acid (HNO$_3$ for Trace Metal Analysis, Baker Instra-Analyzed Plus, Assay 67-70% (w/w)) and 1 mL of hydrochloric acid (HCl for Trace Metals Analysis, Baker Instra-Analyzed Plus, Assay 34-37% (w/w)) were applied to each sample. Lesser volumes of acid were used compared to the EPA method due to the much smaller amount of mineral dust in a sample compared to soil and sediment material typically digested using the method. The mixture was left to predigest the filter for at least 30 minutes, at which point the digestion vessel was sealed and placed into the sample carousel of a Microwave Digestion Apparatus (CEM, Mars Xpress, Model MARS 230/60). A pre-programmed EPA 3051 method available with the instrument was utilized, and so the Microwave Digestion Apparatus was programmed to ramp the temperature of the solution to 175 °C over 5.5 minutes and then hold at this temperature for 4.5 minutes, ending with a 15-minute cool-down period. Once the digested samples were cooled, the vessels were opened carefully in a fume hood to release any remaining pressure. To obtain an acid concentration suitable for passing through a Nylon syringe filter (VWR® Syringe Filter, Polypropylene housing, 25 mm diameter, Nylon membrane, 0.45 μm pore size), 20 mL of Type-1 water was added to each digestion. The samples were filtered into tared 50-mL polypropylene centrifuge tubes that were either previously acid washed or were certified metal-free and given an initial rinse with Type-1 water. Type-1 water was used to thoroughly rinse the vessels and then passed through the syringe filter multiple times for optimal recovery. The centrifuge tubes were then completed to about 45 mL with Type-1 water and then reweighed to determine the final volume gravimetrically.

**2.3 Elemental analysis by ICP-MS.** External calibration standards were prepared using a multi-element ICP-MS standard (Inorganic Ventures, IV-ICPMS-71A, ICP-MS Complete Standard – 1% HNO$_3$) between 1 and 100 μg L$^{-1}$ for major elements and between 0.01 and 5 μg L$^{-1}$ for minor elements. Quality Control Standards were prepared using a separate multi-elemental ICP-MS standard (High-Purity Standards, ICP-MSCS-PE3-A, High Purity Standards Solution A of ICP-MS PE Calibration Standard 3 in 5% HNO$_3$), at concentrations representing both the center and lower range of each calibration range. Sc, Y, and Tb were used as internal standards, introduced in-line prior to the instrument spray chamber. A Nexion 5000 Multi-Quadrupole ICP-MS (Perkin-Elmer®) was used for the analysis. O$_2$ was used as a reaction gas to reduce interferences for the As, Cr, Cu, Fe, and V isotopes of interest. Otherwise, elements were analyzed in standard quadrupole mode. The details of the ICP-MS instrumental parameters can be found in Table S1. The recovery of analytes was determined by preparing replicates of Standard Reference Material (SRM) 2710a Montana I Soil, as its matrix most closely resembles that of mineral





dust, although its size distribution is coarser. Recovery data, along with the method LOD, is presented in Table S2. Approximately 10 mg of the SRM was digested along with a quartz fiber sampling filter half, so to best reproduce the digestion environment that the samples are subject to. The low recovery of some major elements in the SRM could be due to incomplete

digestion of the silicate structures found in soil particles, which could be explained in part by the non-use of HF in the digestion, which is normally responsible for breaking down these structures (Chen and Ma, 2001). The obtained recoveries are similar to those presented for the "Leachable concentrations Determined Using USEPA Method 200.7 and 3050B" for this SRM, which are precursor methods to the 3051A method adapted and employed here (Gonzalez and Choquette, 2018). The elemental concentrations presented in this work have been adjusted using the obtained recoveries (Linsinger, 2008).


**2.4 Size Distribution Determination.** Two Multichannel Optical Particle Counters (OPC; Fast-Response Multichannel Monitor, FAI Instruments) were used to measure the size distribution and concentrations of the PM during the campaign. The OPC inlet heights were matched to the ARA N-FRM sampler inlet heights. Particles of diameters between 0.28 and 10 μm were measured at an initial time resolution of 4 Hz and a flow rate of 1 L min$^{-1}$. The instrument pumped filtered air at a rate

of 6 L min$^{-1}$ in line with the sampled air to yield a dilution factor of 6:1 to avoid saturation of the detector during highly dusty periods. Raw OPC data is made accessible via a repository (Downey, 2025c). OPC data from 6 m above the ground was not used in this manuscript, as this OPC did not demonstrate good agreement with the corresponding samples collected gravimetrically, indicating that it was possibly not well-calibrated for accurate particle counting.

The dust size distribution for PM$_{10}$ was also measured offline. Near the end of the campaign between June 28th and

July 5th, Nucleopore membrane filters were used to sample PM$_{10}$ for 24-hour periods using the same ARA N-FRM Samplers as above but operating at a flow rate of 10.0 L min$^{-1}$. All Nucleopore filters (lab blank, field blank, and samples) were cut in half and weighed using a balance (Mettler Toledo XSE105), with the filter half to be extracted weighed directly in a 50-mL centrifuge tube. 5 mL of Type-1 water was added to each tube containing a filter half. The tubes were then secured to an orbital shaker and set to run at 200 rpm for 60 minutes at room temperature. The filter halves were removed from the tubes

and the remaining particle suspensions were stored at −20 °C until size distribution measurements were performed. Size distributions were measured using a Beckman Coulter Multisizer 4e Particle Analyzer, equipped with a 30 μm aperture tube.

**2.5 Meteorological Measurements.** Wind speed and direction was measured throughout the campaign using cup anemometers equipped with wind vanes (NRG 40C) at various heights between ground level and 10 m, averaged to 10-minute

resolution. The data used for the purposes of this publication was measured by an anemometer positioned between 2.5 and 4.7 m above ground during the reported measurement period. The height of the anemometer was adjusted at times to account



for sand dunes migrating in the sampling site. Temperature and RH were measured throughout the campaign using digital air temperature and RH sensors (Campbell CS215) at 1.3 m off the ground.

### 3. Results and discussion

**3.1 Gravimetric Analysis of Mineral Dust Resolved to Size Class.** Daily mass concentrations were highly variable during the measurement period. Figure 3 shows a time series of the PM mass concentrations measured gravimetrically, on a 24-hour basis for TSP and $PM_{10}$, and on a 48-hour basis for $PM_{2.5}$. The corresponding recommended daily limits by the WHO for $PM_{10}$ and $PM_{2.5}$ exposure were exceeded in the Valley several times throughout the campaign, 45 and 15 µg m$^{-3}$,

respectively.(World Health Organization, 2021) The PM concentrations for each day are also presented in Table S3. The average daily TSP concentration at 3.3 m off the ground measured during the campaign was $8200 \pm 3900$ µg m$^{-3}$, while that for $PM_{10}$ was $372 \pm 79$ µg m$^{-3}$, and that for $PM_{2.5}$ was $84 \pm 62$ µg m$^{-3}$, where the uncertainties express the standard error of the mean. The maximum daily dust concentrations measured gravimetrically at 3.3 m off the ground for TSP, $PM_{10}$, and $PM_{2.5}$ were $4.61\times10^{4}$, 922, and 327 µg m$^{-3}$, respectively. These maxima occurred near the end of June and signify a peak of dust

activity during the campaign. A clear gradient of dust concentrations with height was determined by comparing measurements at 6.3 m against those taken at 3.3 m above the ground. On average, the ratio of TSP concentration in air at 6.3 m to that at 3.3 m was $0.61 \pm 0.42$, while for $PM_{10}$ it was $0.69 \pm 0.24$, as determined by gravimetric concentration results. On average during the measurement period, nearly a quarter of TSP mass concentration in air could be attributed to the $PM_{10}$ size class, as the portion of $PM_{10}$ in TSP measured during the campaign was $0.23 \pm 0.11$. However, this ratio decreased with an

increasing magnitude of the dust event from 0.5 to almost zero, Fig. S1.

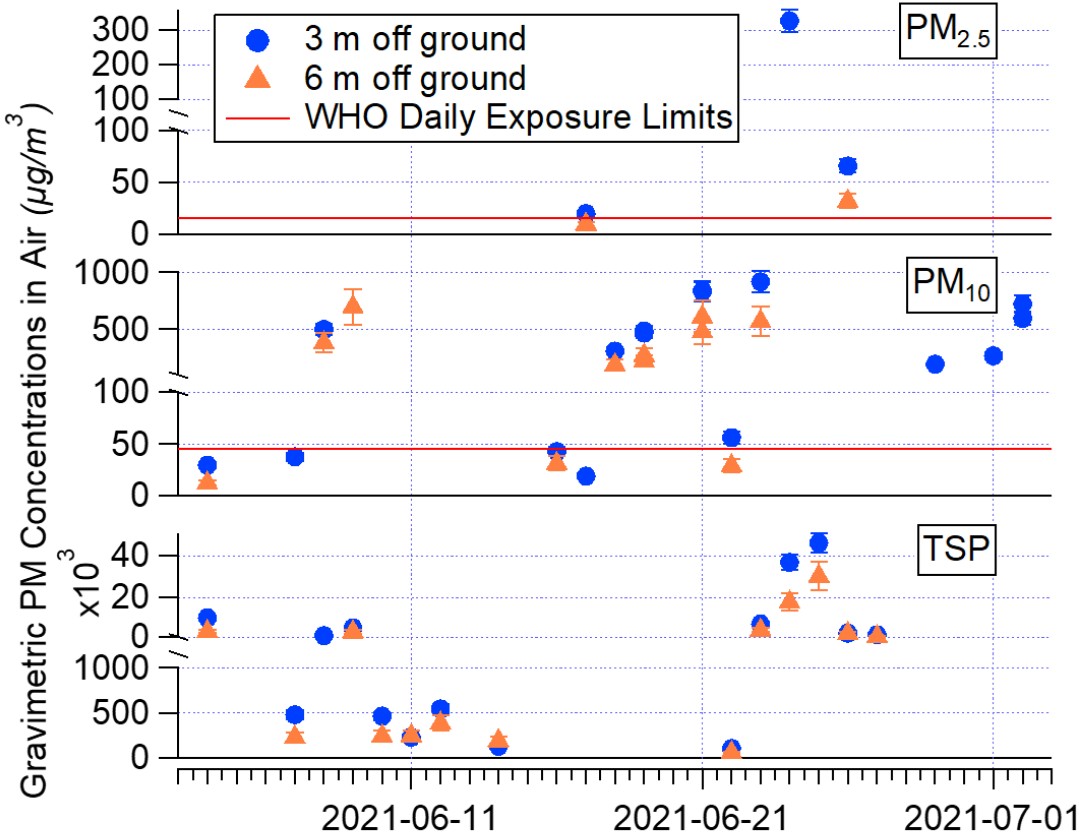

**Figure 3. Gravimetric PM concentrations in air measured during Kluane 2021 campaign. TSP and PM₁₀ samples were collected over 24-hour periods, while PM₂.₅ samples were collected over 48-hour periods. Sampling began within one hour of 9 am local time of the date indicated. Error bars indicate the precision of the measurement (14%), which was estimated by the mean percent difference of multiple duplicate samples collected.**

### 3.2 Dust Concentrations, Size Distribution, and Temporal Variation Analysis from OPC Data with Meteorological Factors

OPC data was collected continuously during the campaign, with some interruptions experienced, Fig. 4. Dust concentration measurements from the OPC at 3.3 m were compared to those obtained gravimetrically from filter sampling and are presented in Fig. 5. Compared to the unity function, the OPC slightly overestimated PM₁₀ on low-concentration days and slightly underestimated it on high-concentration dust days, as indicated by the line of best fit. The average ratio of concentration determined by the OPC method relative to the gravimetric method is 128%. The OPC did not agree well with the gravimetric





method with respect to PM$_{2.5}$ concentrations, Fig. S2, having a ratio of 35%, but the number of datapoints available for this

comparison is very limited.

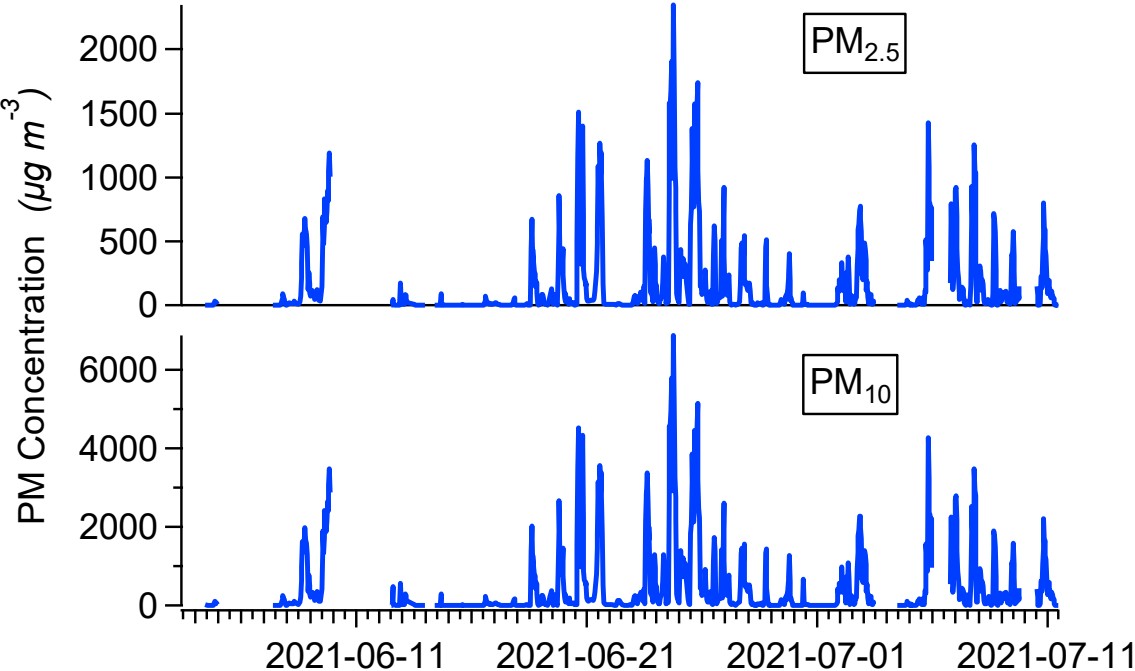

**Figure 4. PM$_{10}$ and PM$_{2.5}$ concentrations measured at Down Valley site in Ä'äy Chù Valley using OPC at 3.3 m above ground level, averaged to 1-hour**




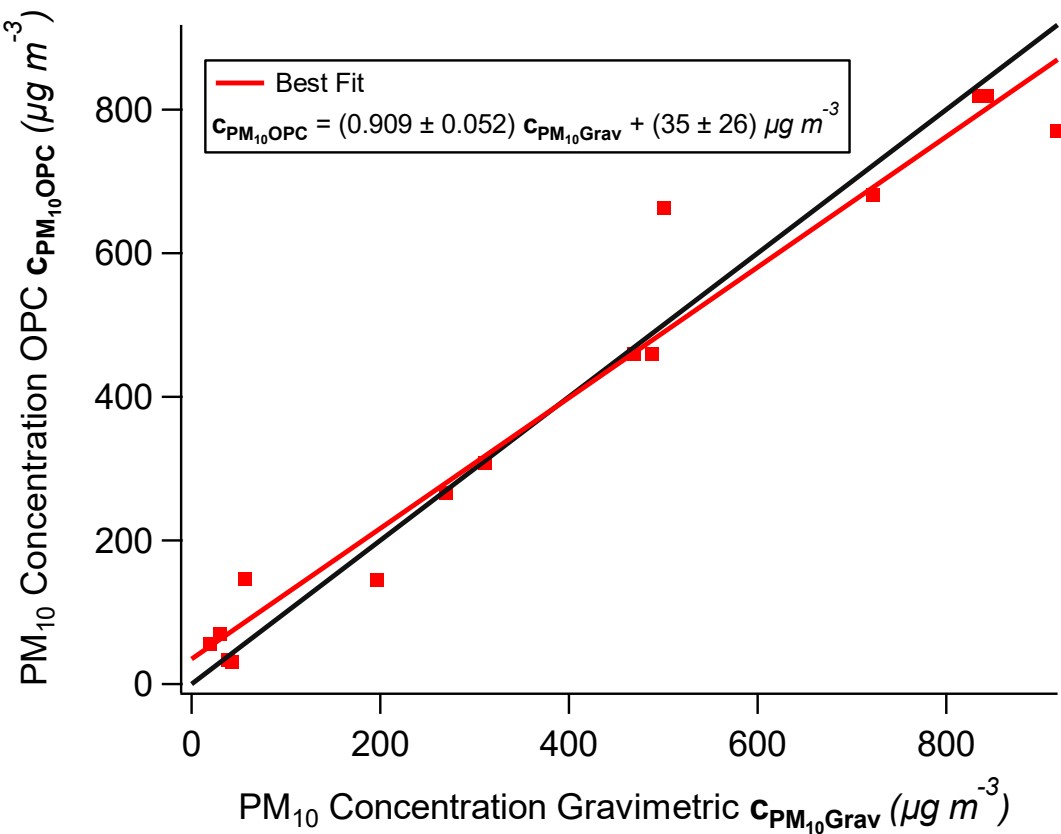

**Figure 5. Comparison of $PM_{10}$ measurements at 3 m using OPC and Gravimetry. The black line represents the unity function. The linear function of best fit is presented along with associated standard errors.**

As shown in Fig. 4, a clear diurnal cycle of dust activity is observed at the sampling site, which is also summarized on an hourly basis by Fig. 6. Tabulated results of the diurnal trends for PM concentrations and meteorological conditions are available in the manuscript assets (Downey, 2025b). The diurnal trend matches visual observations made in the field. Often, mornings exhibited little-to-no dust activity, with more pronounced extreme dust activity occurring in the afternoons. As such, the lowest PM concentrations occurred at 5am on average while the daily maximum in dust activity tended to occur at 7pm local time, which boasted an average $PM_{10}$ and $PM_{2.5}$ concentration of $1378 \pm 34$ µg m$^{-3}$ and $467 \pm 12$ µg m$^{-3}$, respectively. This maximum corresponds with meteorological conditions, Fig. 7, as wind speeds tended to be highest at this time of day on average, $6.47 \pm 0.30$ m s$^{-1}$. There is also a correspondence between the afternoon increase of the 75[th] percentile wind speeds and PM concentration at 2pm local time, with a 1.3-fold increase in wind speed corresponding to a 6-fold increase in dust concentrations for both size fractions, compared to the previous hour. Wind direction, Fig. S3, largely followed the contours



of the valley over the sampling period, Fig. 1. The maximum dust concentrations lag an hour behind the maximum average air temperature and two hours behind the minimum relative humidity during the campaign, 18.28 ± 0.47 °C and 36.2 ± 1.6 %, respectively; suggesting the warm and dry air conditions prime sediment of the ground for optimal dust emission. It has

often been suggested that the primary driving factor of the high-speed winds causing the dust emission in the 'A'ay̆ Chù' Valley was the phenomenon of katabatic winds derived from nearby glaciers (Denton and Stuiver, 1967; Bachelder et al., 2020). However, a more thorough investigation into this phenomenon in the 'A'ay̆ Chù' Valley has revealed that this is most likely not the case, and that topographically channeled flows have been misclassified as katabatic winds in these cases. A study on the forcing mechanisms of strong surface winds in this valley over the period of July 2021 to September 2022

revealed that 50.0% of high-wind-speed events occurred under high above-valley wind speeds; while a summertime valley wind system is likely responsible for persistent nocturnal high-speed winds near the valley delta, contributing to 28% of summer high-speed winds under distinctly calm conditions aloft (Bellamy et al., 2025b).





**Figure 6. Hourly diurnal variation in PM$_{10}$ and PM$_{2.5}$ concentrations. Black crosses indicate the mean, while the center**

**line of each box indicates the median, with each box edge indicating the 25 and 75 percentile ranges. Whiskers indicate**

**the 5 and 95 percentile ranges.**







**Figure 7. Hourly diurnal variation in wind speed, air temperature, and relative humidity. Black crosses indicate the mean, while the center of each box indicates the median, with each box edge indicating the 25 and 75 percentile ranges. Whiskers indicate the 5 and 95 percentile ranges. Meteorological factors were measured the campaign between June 2nd and July 12th, 2021.**



The number, surface area, and volume distribution of the mineral dust measured during the campaign by the OPCs is depicted in Fig. 8 and summarized in Table 1. As shown by Fig. S4, the average size characteristics of the dust do not differ substantially when compared to that for the top 10th percentile of dust concentrations during the campaign. Mean particle diameters obtained

from the Coulter Counter analysis were somewhat smaller compared to the OPC, when constraining both techniques to the size ranges they have in common (0.6 – 10 μm), Fig. 9 and Table 2. As discussed, the assumption of sphericity does not have a straightforward, directional effect on the assumed particle size of mineral dust when measured with OPC (Collins et al., 2000; Knippertz and Stuut, 2014). The compositional difference of mineral dust compared to the polystyrene latex spheres (PLS) used to calibrate the OPC is not expected to make a substantial difference either, as the refractive index of PLS  is

within the range of refractive indices for the clay minerals largely composing this fraction, which is between 1.47 and 1.68.(Bachelder et al., 2020; Mukherjee, 2013; Smart and Willis, 1967) Alternatively, the mineral dust particles may contain some soluble material, which is dissolved in the electrolyte solution during the Coulter Counter analysis leading to a low bias in the particle size measurements. In fact, it was previously surmised that the $PM_{10}$ mineral dust particles in this Valley were largely comprised of silt-sized clay mineral aggregates (Bachelder et al., 2020). It is therefore possible that these aggregates

disaggregate upon introduction to the electrolyte solution used for Could Counter analysis, resulting in the perceived shift in size distribution.  The measured particle size distribution is coarser than that measured at the same location and height above the ground in 2018 by Bachelder et al, whose OPC results yield a particle mass distribution with a mean diameter of 3.17 μm and a variance of 4.18 $μm^2$ (Bachelder et al., 2020). This could be due to the depletion of finer material in the sediment over the years between campaigns.






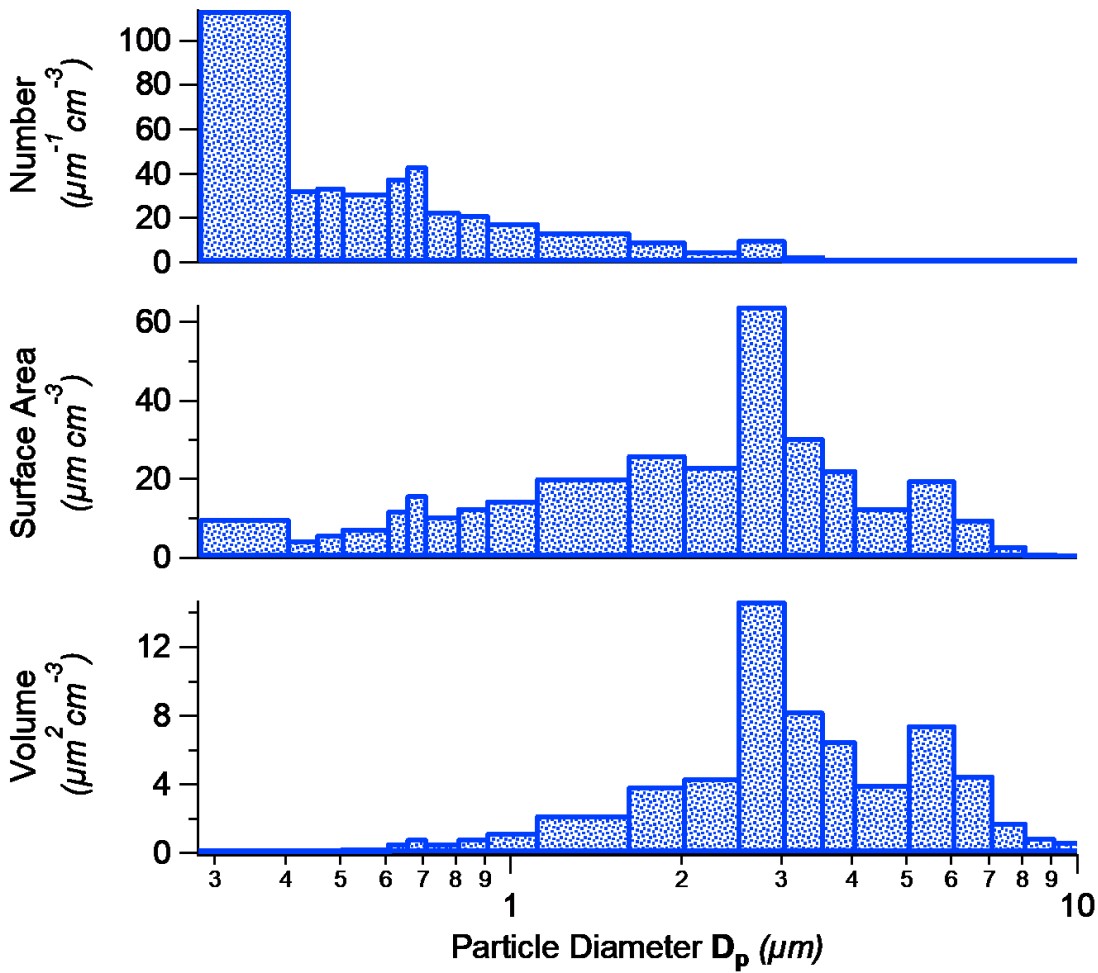

**Figure 8. Aerosol number, surface, and volume distributions measured at Down Valley site in 'A'ạy Chù' Valley using OPC at 3.3 m above ground level. Values are normalized to bin width.**






**Table 1. Moments of the size distributions of PM measured by OPC at 3.3 m above ground during Kluane 2021 Campaign.**

| Distribution | Number | Surface Area | Volume 365 |
|---|---|---|---|
| **Total concentration** | 58.2 cm$^{-3}$ | 151 μm$^2$ cm$^{-3}$ | 40.7 μm$^3$ cm$^{-3}$ |
| **Number Mean Diameter $\overline{D_p}$ (μm)** | 1.43 | 3.56 | 4.43 |
| **Variance $\sigma$ (μm$^2$)** | 1.78 | 3.53 | 4.08 |

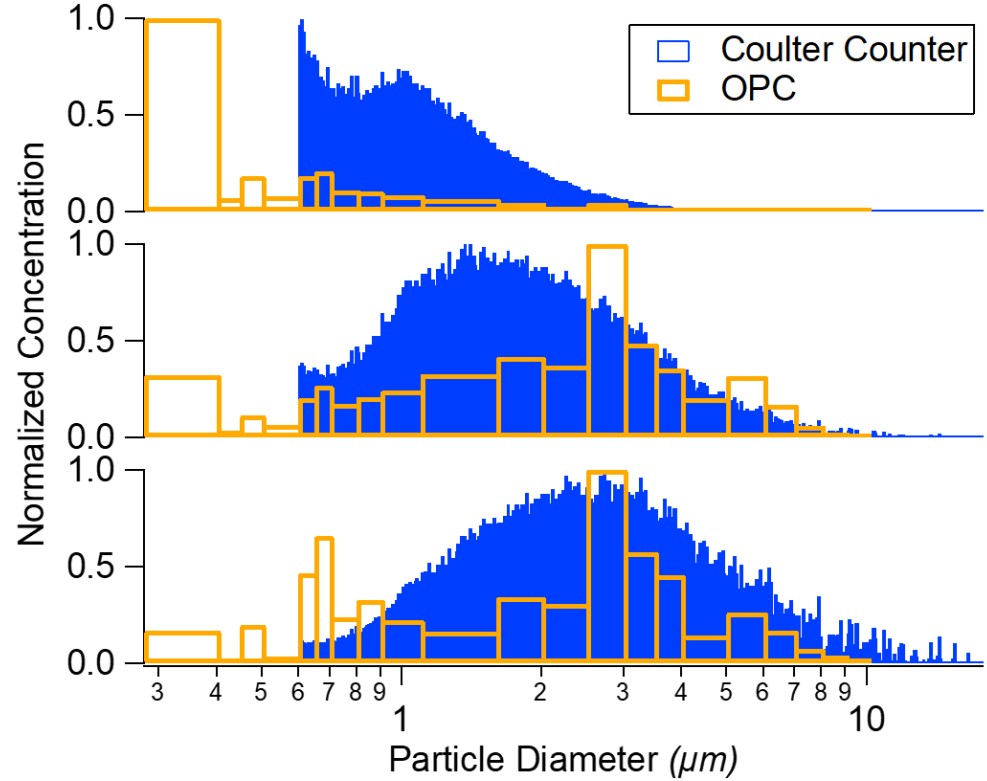


**Figure 9. Normalized a) number, b) surface area, and c) volume concentration distributions measured simultaneously at Down Valley site in 'A'äy Chù' Valley using OPC and with Coulter Counter at 3.3 m above ground, normalized to bin widths**





**Table 2. Comparing moments of particle size distributions measured by the OPC and Coulter Counter during Kluane 2021 campaign for full range and common range of each method**

| Parameter | | OPC | | Coulter Counter | |
| :---: | :---: | :---: | :---: | :---: | :---: |
| | | Full Range | Common Range | Full Range | Common Range |
| Mean Diameter (µm) (Variance of distribution (µm$^2$)) | Number | 1.21 (1.56) | 1.96 (1.81) | 1.52 (0.85) | 1.51 (0.82) |
| | Surface Area | 3.56 (3.68) | 3.64 (3.50) | 2.99 (4.02) | 2.87 (3.01) |
| | Volume | 4.57 (4.64) | 4.48 (4.12) | 4.34 (7.49) | 3.92 (4.32) |

**3.3 Elemental Concentrations**. A subset of the collected filter samples with the greatest PM masses for each size class were used to determine the mass concentration of several metal and metalloids present in the mineral dust by ICP-MS. The mass concentrations of various metal(loid)s in the mineral dust of the 'A'äy Chù' Valley had been previously determined for the PM$_{10}$ size class in a previous study (Bachelder et al., 2020). This previous analysis covered As, Cd, Co, Cs, Cu, Mn, Ni, and Pb in PM$_{10}$, as well as in source material, both bulk and fine (less than 53 µm diameter) soil. Notably, this previous work found that the determined metal(loid) concentrations increased with finer grain sizes from bulk to fine soil, to PM$_{10}$. In the present work, we have further investigated this trend by comparing three PM size classes (TSP, PM$_{10}$, and PM$_{2.5}$) and by expanding the number of elements analyzed. The trend of increasing trace element concentrations with decreasing size fraction is observed for many elements in samples collected during this campaign, Fig. 10. The enrichment of elements in the PM$_{10}$ and PM$_{2.5}$ size classes are summarized in Table 3, where enrichment in this case is the ratio of the concentration of a given element by mass in one size fraction to that of another size fraction. The enrichment of metal(loid)s with finer grain sizes is consistent with findings made in the field of glacial till prospecting for mining purposes, which analyzes fine sediments resulting from glacial comminution processes to aid in the identification of mineral sources (Shilts, 1993). Shilts et al. observed that in glacial sediments many trace metal(loid)s present in the sediments increase substantially in concentration with finer size fractions, with the highest concentrations found in the primarily clay-sized fraction of <4 µm (Shilts, 1984a). This previous work proposed that these elements are present within the structure of phyllosilicates or scavenged by secondary oxides. Both phases occur preferentially among particles finer than about 10 µm, which could explain the enrichment observed for metal(loid)s in the fine dust size fractions here. It is also suspected that weathering processes play a role in the enhancement



of metal(loid) concentrations. Labile minerals, such as sulphides, are weathered within and below the postglacial solum, which

is the surface or subsoil layers that have undergone the same soil-forming conditions following a glaciation event. This

weathering is accompanied by an increase in metal(loid) concentration in the clay-sized fraction near the surface. This fraction

possesses a high specific surface area and a high ion exchange capacity, and is capable of adsorbing and incorporating liberated

metal(loid)s from surrounding sources (Shilts, 1993). This enrichment effect should be taken into consideration when

assessing the potential health effects from respirable mineral dust due to metal(loid) exposure. In comparing metal(loid)

concentration results in $PM_{10}$ with those determined by Bachelder et al. in 2018 at the same location and taking care to account

for differences in recovery of the methods used, performing Welch's t-tests to 95% confidence between the datasets reveals

that the dust analyzed in the present study contains significantly less As, Cd, Cu, Fe, Mn, and Ni on a per-mass basis, Table

4. The $PM_{10}$ analyzed in the present study contained between 14 and 49% less of these elements, while both studies yielded

similar results for Co, around 25-26 $\mu g\ g^{-1}$. Pb appeared somewhat depleted but did not yield a significant difference. The

relative depletion of these elements over the three-year period between the field studies may be related to the observed

difference in particle size distribution, Fig. 11. As discussed, the finer fractions of mineral dust contain greater concentrations

of many metal(loid)s. Due to the greater mobility of the finer particles, it is therefore possible that the dust has become coarser

and less abundant in clay-sized minerals compared to 2018 resulting in the lower observed concentrations of trace elements.

As noted in Shiltz (1993), the content of metal(loid)s in glacial till is greatest below the 4 $\mu m$ diameter. Correspondingly,

50.1% of the volume distribution of $PM_{10}$ measured in this campaign falls below this value compared to 81.4% of the

distribution measured in the 2018 campaign. Furthermore, recent single-particle ICP-MS work has revealed that sub-micron

mineral dust particles at the Down Valley site are largely dominated by Fe, suggesting that Fe-containing particles are more

susceptible to erosion, which is consistent with the relatively large decrease in Fe concentration between 2018 and 2021

(Tardif et al., 2025).





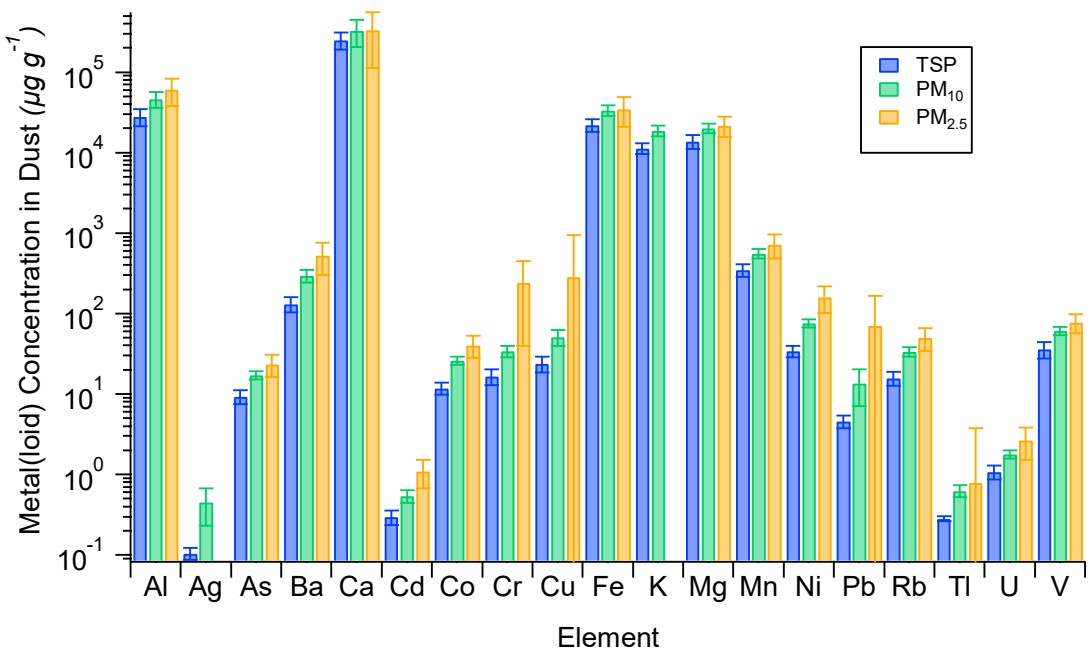

**Figure 10. Average metal(loid) concentrations measured in mineral dust PM collected during 2021 Kluane campaign with associated standard uncertainty at 95% confidence.**







**Table 3. Metal(loid) enrichment by mass concentration in 'A'ậy Chù' Valley mineral dust by size class, with associated propagated uncertainty at 95% confidence**

|  | $PM_{10}/TSP$ | ± | $PM_{2.5}/TSP$ | ± | $PM_{2.5}/PM_{10}$ | ± |
|---|---|---|---|---|---|---|
| Mg | 1.48 | 0.21 | 1.37 | 0.13 | 0.93 | 0.10 |
| Al | 1.93 | 0.27 | 2.60 | 0.33 | 1.35 | 0.18 |
| K | 1.73 | 0.17 |  |  |  |  |
| Ca | 1.39 | 0.21 | 1.43 | 0.14 | 1.02 | 0.13 |
| V | 2.08 | 0.23 | 2.77 | 0.45 | 1.33 | 0.20 |
| Cr | 2.33 | 0.31 | 17.0 | 9.5 | 7.3 | 4.1 |
| Mn | 1.95 | 0.20 | 2.43 | 0.43 | 1.25 | 0.21 |
| Fe | 1.47 | 0.22 | 1.51 | 0.24 | 1.02 | 0.17 |
| Co | 2.54 | 0.27 | 3.58 | 0.75 | 1.41 | 0.28 |
| Ni | 2.37 | 0.26 | 3.60 | 0.60 | 1.52 | 0.23 |
| Cu | 1.98 | 0.23 | 15 | 12 | 7.4 | 6.0 |
| As | 2.11 | 0.25 | 3.22 | 0.59 | 1.53 | 0.26 |
| Rb | 2.44 | 0.26 | 3.46 | 0.78 | 1.42 | 0.31 |
| Ag | 4.3 | 1.0 |  |  |  |  |
| Cd | 2.11 | 0.26 | 3.38 | 0.52 | 1.60 | 0.25 |
| Ba | 2.67 | 0.34 | 4.25 | 1.23 | 1.59 | 0.45 |
| Tl | 2.28 | 0.18 | 2.81 | 0.83 | 1.23 | 0.37 |
| Pb | 2.32 | 0.26 | 4.16 | 0.65 | 1.79 | 0.27 |
| U | 1.89 | 0.21 | 2.46 | 0.69 | 1.30 | 0.35 |




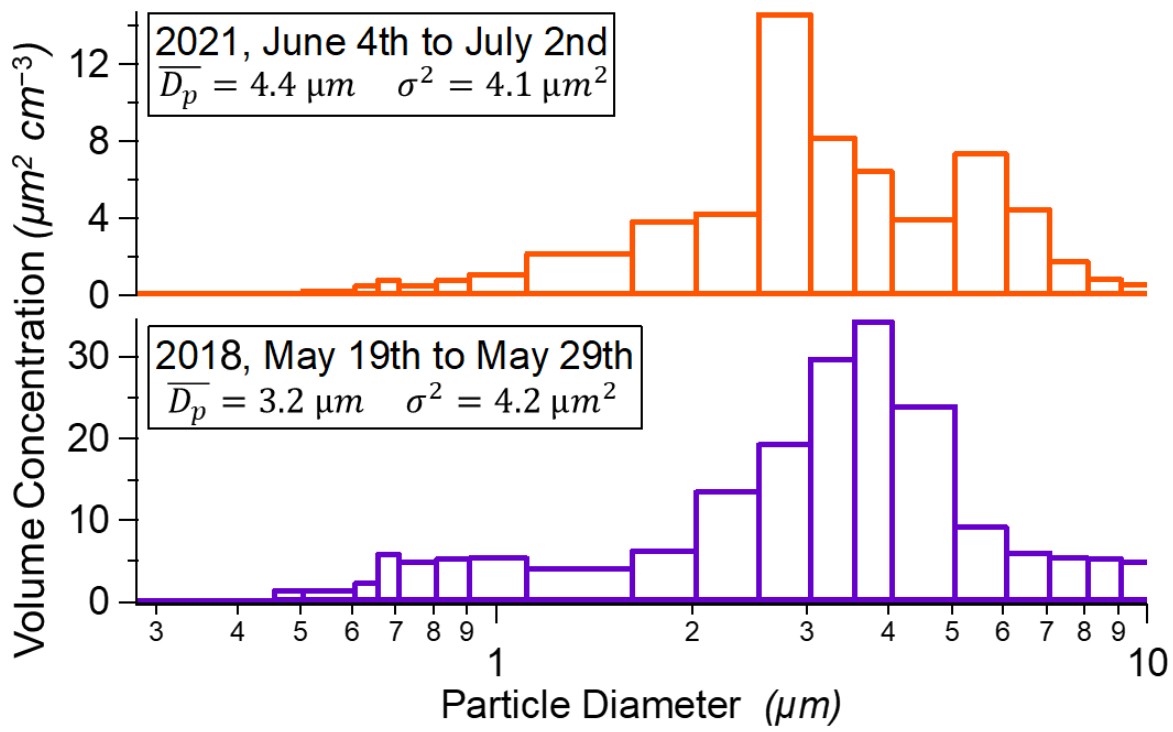

**Figure 11. Bin width-normalized particle volume distribution comparisons of Kluane mineral dust between 2021 and 2018 at 3.3 m and 3.5 m above the ground, respectively. 2018 distribution adapted from Bachelder (2020).**






**Table 4. Average concentrations of elements in Kluane mineral dust PM$_{10}$ with expanded uncertainty to 95% confidence comparison between present 2021 campaign and previous 2018 campaign (Bachelder 2020)**

| Element | 2021 Average Concentration in PM$_{10}$ with expanded uncertainty (µg g$^{-1}$) | N | 2018 Average Concentration in PM$_{10}$ with expanded uncertainty (µg/g$^{-1}$) | N | Equal (Welch's) |
|---|---|---|---|---|---|
| As | 17.1 ± 2.1 | 22 | 20.7 ± 1.1 | 9 | N |
| Cd | 0.542 ± 0.094 | 19 | 1.07 ± 0.30 | 9 | N |
| Co | 26.2 ± 3.2 | 22 | 25.1 ± 1.0 | 9 | Y |
| Cu | 51 ± 11 | 5 | 74.0 ± 3.8 | 9 | N |
| Fe | 33'100 ± 4'800 | 17 | 51'800 ± 2'200 | 9 | N |
| Mn | 553 ± 75 | 22 | 786 ± 32 | 9 | N |
| Ni | 75.7 ± 9.1 | 20 | 87.6 ± 3.7 | 9 | N |
| Pb | 13.7 ± 6.6 | 13 | 16.45 ± 0.97 | 9 | Y |

Metal(loid) concentrations in air are presented for each applicable sample in manuscript assets (Downey, 2025a). The Yukon has 8-hour and 15-minute exposure limits for various metals in air as part of their Workplace Health Regulations (Yukon

Regulations, 1986), and to our knowledge, does not have similar standards for ambient air on a 24-hour basis. Therefore, standards established by the Province of Ontario Ministry of the Environment are used for comparison on a 24-hour basis.(Human Toxicology and Air Standards Section, 2020) Ambient exposure to several metals in air exceeded the Ontario Ministry of the Environment standards multiple times throughout the campaign, for TSP (As, Co, Cr, Fe, Mn, Ni, V), PM$_{10}$ (Fe, Mn), and PM$_{2.5}$ (Fe, Mn). It is important to note that the sampling location is right at the dust source, so these results

represent the associated maxima. The concentrations of PM and associated metal(loid)s decreases with distance from the valley. Therefore, a study of the horizontal gradients of the metal(loid) concentrations in the region would be beneficial for understanding the potential health impacts on the public. Furthermore, the bio-accessibility of these metal(loid)s relative to other common respirable PM types would be an important factor to consider, as the entire metal(loid) content likely does not fully dissolve in lung fluid before expulsion via bodily pathways (Kastury et al., 2017; Olumayede et al., 2021). Determining



the bioaccessibility of metal(loid)s in respirable PM is an active area of research, with a dominant method still not fully

established (Kastury et al., 2018).

### 4. Conclusion

Metal and metalloid content in PM is known to modulate the impact it has on air quality, as respiration of some metals at

elevated concentrations may lead to negative health outcomes (Chen et al., 2023; Badaloni et al., 2017; Liu et al., 2025).

Furthermore, $PM_{2.5}$ has a measurably greater impact on health on a per-mass basis than $PM_{10}$, and in turn $PM_{10}$ has a greater

impact than TSP, given the progressively greater propensity to penetrate deeply into the respiratory tract. The results presented

here suggest that mineral dust in the $PM_{2.5}$ and $PM_{10}$ size class possess greater metal(loid) concentrations on a per-mass basis,

further contributing to their potential health effects upon respiration. For instance, the concentration of As in $PM_{10}$ was 2.11

$\pm$ 0.25 times higher than that present in the TSP size fraction, and that of $PM_{2.5}$ was 1.53 $\pm$ 0.26 times higher than in the $PM_{10}$

size fraction. A similar trend was noted for several other metals that are known to be problematic in PM exposure, such as

Pb, Mn, and Cr (Liu et al., 2025). In contrast, these enhancements were not observed or were relatively small for some major

elements (e.g., Mg, Ca, and Fe). This could be due to glacial comminution leading to the production of fine glacial sediment

material that is rich in the clay-sized particle fraction that is known to possess metallic enrichments. and which is more

susceptible to aeolian erosion that produces $PM_{10}$ and $PM_{2.5}$. Thus, such sources present a compounded risk for human health

effects associated with dust exposure. With the relatively recent emergence of this particular dust source in the Kluane region,

and the likelihood of similar glacial sediment dust sources emerging in other high-latitude locations by similar mechanisms

of glacial recession due to climate change (2021), attention must be paid to the potential for local populations to be exposed

to elevated metal(loid) concentrations in ambient air. This region in particular is home to the Champagne, Aishihik, and

Kluane First Nations (Neufeld, 1972), who are impacted by the local dust emissions as well as other environmental impacts

of climate change, which is more rapid in northern regions.

The size distribution properties of the mineral dust were characterized in this study using online OPC and offline

Coulter Counting techniques for validation and comparison purposes. The volume distribution of the dust measured using the

OPC yielded a mean diameter of 4.48 μm, with a variance of 4.12 μm$^2$, while that measured by the Coulter Counter was 3.92

μm with a variance of 4.32 μm$^2$, calculated using the common ranges of the instruments. The two methods provide similar

results with respect to the mean diameter, with the slightly lower diameter determined by the Couter Counter being attributable

to the disaggregation of particles when they are dispersed in the electrolyte solution before analysis. The size distribution



measured in this campaign was coarser than that also measured by OPC in 2018, which corresponds with the decrease in metal(loid) concentrations in the dust between these campaigns.

A more detailed study of the extent of these mineral dust emissions, whether by means of modelling or additional sampling campaigns, would be beneficial to obtaining a broader understanding of high-latitude mineral dust size and compositional properties. Additionally, identifying proglacial valleys in other regions where glacial recession is occurring and conducting field campaigns with similar goals would help affirm the findings contained here.

**Data availability**

Data corresponding to this manuscript has been included herein or is available at FAIR-aligned data repository Borealis and is cited with the appropriate DOI. The authors are willing to provide additional information for data that may be of interest to readers upon request.

**Author contribution**

PLH and JK supervised this project and provided resources. ARD, JK and PLH administered funding acquisition. ARD, DB, JK, PLH contributed to project conceptualization and campaign planning, with ARD developing the methodology. ARD and DB conducted the field investigation. ARD and AD performed the formal analysis and data validation. ARD prepared the figures and wrote the initial manuscript draft. All authors contributed to manuscript revisions.

**Competing interests**

The authors declare that they have no conflict of interests.

**Acknowledgements.**

We acknowledge support from the Natural Science and Engineering Research Council of Canada (Discovery Grants RGPIN-05002-2014 and RGPIN-2016-05417 for Patrick Hayes and James King, respectively), the Canada Foundation for Innovation (Leaders Opportunity Fund Projects 32277 and 36564 for Patrick Hayes and James King, respectively), and the Canadian Mountain Network, a Canadian Government Network of Centers of Excellence (PV143493-NCE). ARD acknowledges support from the Polar Knowledge Canada Northern Scientific Training Program (NSTP). ARD also acknowledges scholarships from the *Centre de recherche en écotoxicologie du Québec* (EcotoQ), a strategic cluster funded by the *Fonds de recherche du Québec – Nature et technologies*. ARD acknowledges logistical and equipment support from Dr. Kevin Wilkinson and Madjid Hadioui for the ICP-MS analysis. ARD and DB acknowledge the hospitality and service provided by the Outpost Research Station. We acknowledge that the sample collection in this work was done on the traditional territories of the Kluane First Nation and the Champagne and Aishihik First Nations, as well as on the White River First Nation territories, with their permissions obtained through a sampling permit (21-35S&E) as required by the Yukon. We also



acknowledge Parks Canada for providing permissions for sampling in the Kluane National Park and Reserve (permit KLU-2021-38964).

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
