# Peer review of "Emerging Mineral Dust Source in 'A'äy Chù' Valley, Yukon, Canada Poses Potential Health Risk via Exposure to Metal and Metalloids Enriched in PM10 and PM2.5 Size Fractions"

_EGUsphere, 2025_

## Author Comment (AC1)

**Responses to referee comments on "Emerging mineral dust source…" EGUsphere 2025**

NB: The referee comments are given in plain text, our responses are in **bold text**, and changes to the manuscript are in ***bold italicized text***.

Reviewer 1, Ian Burke:

This study investigates the emergence of a high-latitude mineral dust source in the 'A'äy Chù' Valley, Yukon, Canada, following climate-driven glacial recession. The authors focus on the size distribution, elemental composition, and health implications of particulate matter (PM), particularly $PM_{10}$ and $PM_{2.5}$, enriched with PTEs. The work builds on previous campaigns and expands the scope of elemental analysis. The study addresses a newly intensified dust source in a high-latitude region, which is underrepresented in global dust budgets but disproportionately affects snow albedo and local air quality, linking climate change, glacial recession, and public health. Dual instrumentation (Optical Particle Counter and Coulter Counter) provides cross-validation of particle size distributions. Elemental analysis is well-detailed, including digestion protocols, calibration, and recovery corrections. The study also includes some diurnal variation analysis and comparisons with WHO and Ontario air quality standards.

The study is a nice case study in one valley, it would be interesting to see how the authors would scale up these implications to the wider regional impacts (is this just a health impact in one valley?). How many similar sites have emerged due to glacial recession. Can this data be reflected in dispersion models to elucidate local / regional impacts.

**Firstly, we would like to thank the referee for their helpful comments. We have responded to each comment individually below.**

**This is a great suggestion to consider the wider regional impacts. Some work on dispersion modelling for this site has been attempted (https://ui.adsabs.harvard.edu/abs/2024AMS...10433484N/abstract), but a full study has not been published to date. The main issue for dispersion modelling is resolving accurate wind fields in complex terrain. This valley's sidewalls are steep and currently no meteorological product is available that can accurately resolve surface wind conditions. Trajectory analysis would subsequently be a guess at best and likely fail to include key transport processes governing the dispersion of material.**

**Within Canada, similar sites emitting high-latitude dust have been identified near Lake Hazen (Ranjbar et al., 2021) and other locations (Sayedain et al., 2024).**

Air quality standards are normally considered for urban and industrial dust sources – do these natural source dusts contain materials that are similarly bioaccessible in e.g. oral or lung fluids.

**We are currently applying an inhalation-ingestion bioassay procedure (Kastury et al., 2018) to these samples along with urban samples for comparison and hope to submit this work for publication in the coming months.**

Minor comments

L114 – check if 'Mcmurry' should be 'McMurry'. (again at L125 Mctainsh – best check all Scottish names cited).

**We have checked and corrected all the names.**

L275 Typo - 'respectively.(World Health Organization, 2021)'

**Corrected.**

L346 Typo 'and 1.68.(Bachelder et al., 2020; Mukherjee, 2013; Smart and Willis, 1967)'

**Corrected.**

Fig. 8 – check the units of number, SA and volume, seems odd to me (not an atmospheric scientist), but they don't match Table1 either.

**The presentation of units in figure 8 is intended as-is. The values on the size distribution graph are normalized to the widths of each bin. For example, the initial values for each bin of the Number Concentration distribution in units of [$cm^{-3}$] are divided by the bin widths in units of [μm], making the units for the ordinate of the final graph [$μm^{-1}\ cm^{-3}$]. This is so the area below the curve is proportional to the number concentration of particles, and so that larger bins are not visually over-represented when comparing bins of different sizes. If you would like more details, we took this approach by following section 8.1 of "Atmospheric Chemistry and Physics: From Air Pollution to Climate Change, Third Edition. John H. Seinfeld and Spyros N. Pandis. John Wiley & Sons, Inc. 2016"**

Table 2 – why is Variance of distribution in brackets

**We have chosen to present the mean diameters and variances of the distributions in this way to avoid cluttering the table with too many column headers.**

L389 – 'Shilts et al.' needs a date.

**Corrected.**

L447 Typo - basis.(Human Toxicology and Air Standards Section, 2020)

**Corrected.**

Table 3 - enrichment of PTEs in the fine fraction material is commonly reported – it would be good to see the total enrichment versus the parent material (sediments) so that it could be potentially be predicted the relationship between bulk sediment concentrations and PM concentrations (measurement of sediment concentrations is much easier and much more commonly collected than PM concentrations).

**This is a good prompt. We had soil samples but simply neglected to analyze them. These results have now been added and discussed in the paper. The new figure and text have been copied-and-pasted below for convenience.**

*A clear trend of enrichment emerges in comparing the PM$_{10}$ size fraction to TSP, where significant enrichment is noted for Al, Ag, As, Ba, Cd, Co, Cr, Cu, Fe, K, Mg, Mn, Ni, Pb, Rb, Tl, U, and V. This trend is maintained for all elements when comparing PM$_{2.5}$ to TSP, except for Ag, Cu, Fe, Pb, and Tl, largely due to uncertainties being higher with fewer PM$_{2.5}$ sample results. Further enrichment is noted between the PM$_{2.5}$ and PM$_{10}$ size fractions for Cd, Co, Cr, and Ni, indicating substantial compositional differences between these size classes that are considered respirable. Interesting trends emerge when considering the PM size fractions and the parent soil composition, as select elements, Ag and Mg, exhibit significant depletion (i.e., less in PM$_{2.5}$ and PM$_{10}$ relative to soil). Furthermore, some PM$_{10}$ elements that showed enrichment compared to TSP do not do so in comparison to the parent soil. This could be due to spatial inhomogeneity of the bulk glacial sediment compared to the more uniform mixing of suspended material by winds. As soil sampling was conducted near the DV Site, it is possible that "hot spots" of certain elements may be present in the soil sampled, due to the specific minerals present at a given location. A more spatially broad compositional study of the sediment along the length of the valley would help explore this possibility. It is known that glacial sediments can be compositionally inhomogeneous along their glacial trajectory, owing to the different types of bedrock material that would undergo comminution in different locations, (Rencz and Shilts, 1980). Otherwise, many elements exhibit enrichment like when compared against the TSP size fraction.*

[Figure]

*Figure 10. Average metal(loid) concentrations measured in soil and mineral dust PM collected during 2021 Kluane campaign with associated standard uncertainty at 95% confidence.*

Reviewer 2, Pavla Dagsson Waldhauserova:

This study presents a valuable investigation of high-latitude dust (HLD) observations, focusing on the concentrations of metals and metalloids in airborne dust. It is a rare study that explains the mechanisms and significance of the detrimental impacts of HLD storms on human health. The study provides important evidence supporting the need for exposure limits to potentially toxic particulate matter (PM) components, such as metals and metalloids, each with distinct toxicological properties. The direct relationship between the metallic composition and particle size of HLD has rarely been characterized. This research presents additional breakthrough results by comparing TSP, $PM_{10}$, and $PM_{2.5}$ concentrations at a height of 3.3 m, and by providing dust concentration ratios between 3 and 6 m — offering insights into the vertical dust gradient. Including measurements at 1–2 m height would further strengthen the analysis by capturing the full dust gradient from the surface upwards.

**This is a good idea. We had four PM samplers to work with for this campaign, so we opted to pair them at each of the heights so that we could collect different size fractions simultaneously. However, it would have been interesting to dedicate a portion of the campaign to having them at 4 different heights sampling the same size class, so to see the shape of the gradient with height. We will have to keep this in mind for future campaigns.**

The hourly $PM_{2.5}$ and $PM_{10}$ dust concentrations recorded at Kluane Lake are among the highest measured at high latitudes. The comparison between two measurement methods — optical particle counters (OPC) and gravimetric techniques — with respect to $PM_{10}$ and $PM_{2.5}$ concentrations is also highly valuable. The study offers a clear daily temporal resolution of dust activity and provides a thorough explanation of particle size distributions in this unique environment, allowing comparisons with other HLD sources. It would be useful to highlight whether similar studies exist and add them to the discussion.

**We have added other studies of high-latitude dust and aerosol size distributions to the discussion section (specifically at the end of Section 3.2), so thank you for the prompt. It looks like studies of this kind are quite scarce, especially that directly focus on mineral dust. There are a number of in-situ aerosol size distribution measurements in arctic regions that identify mineral dust as a component, as well as indirect studies of high-latitude mineral dust size distributions (such as by ice cores, snowpack samples, sediment, etc). There are also vertical profiling studies using balloon-mounted instrumentation for size distribution determination that provides a short-term, yet highly extensive measurements of the vertical profile.**

*Despite recent interest in high-latitude dust, direct field-based measurements of size distributions remain scarce, with some examples existing for Icelandic dust (Dupont et al., 2024). The size distributions of source sediments for Icelandic mineral dust have been studied previously as well (Butwin et al., 2020; González-Romero et al., 2024). Mineral dusts derived from snow pack and ice core samples in Greenland, Iceland, and Antarctica have illuminated historical dust properties, including size distribution and composition (Komuro et al., 2024; Albani et al., 2012; Aarons et al., 2017). Icelandic dust size distributions have also been studied by way of balloon-deployed OPC instruments to achieve extensive vertical profiling (Dagsson-Waldhauserova et al., 2019). A similar approach has been taken towards arctic aerosols in the Svalbard archipelago (Porter et al., 2020).*

*Arctic aerosol size distributions measured in Svalbard have received sizable attention in the research community (Moroni et al., 2017; Lai et al., 2025; Rinaldi et al., 2021), and to some extent those measured in Canada as well (Vicente-Luis et al., 2021). While these studies investigate mineral dust as a portion of the mixture of high-latitude aerosols, the present study stands as one of the only in-situ measurements of a high-latitude mineral dust source near ground level.*

Furthermore, this research fills a critical gap in in-situ observations of particle number, surface area, and volume concentrations during active dust storms — particularly in ice-proximal regions. The comparative analysis of mean particle diameters is especially noteworthy. The enrichment of metalloids in finer-sized particles is an important finding for understanding the health impacts of dust exposure.

The paper is clearly written, and the figures effectively support the analyses. I highly recommend this work for publication in EGUsphere, pending minor revisions as outlined below.

**We thank the review very much for this positive feedback.**

Specific comments:

L 78 – Year is missing after the reference

**Corrected.**

L88, 98 – Please delete Meinander et al. before the actual reference in the brackets

**Corrected.**

L100-101 – Maybe the study of Panta et al. (2025) could be stated here:

Panta et al. (2025): Unveiling single-particle composition, size, shape, and mixing state of freshly emitted Icelandic dust via electron microscopy analysis, EGUsphere, https://doi.org/10.5194/egusphere-2025-494, 2025.

**Thank you, we must have missed this article upon writing. The text in the introduction has been updated to reflect this. Panta et al. (2025) does well to characterize some major metallic components of a high-latitude dust source with relation to size, along with other elemental components. The present work of course extends this to include many trace metals as well, so these are complimentary approaches to studying high-latitude dust composition.**

*HLD composition has been characterized to some extent on a size-resolved basis with respect to mineral content (Kandler et al., 2020; Barr et al., 2023), as well as for some major metallic components using Scanning Electron Microscopy (SEM) (Panta et al., 2025). However, the direct relationship*

*between the metallic composition of mineral dusts, including trace metals, and their size class in post-glaciated high-latitude regions has not yet been characterized, to our knowledge*

L134, 275, 346, 447 – Move the dot behind the bracket please

**Corrected.**

L139-141 – There are also specific conditions when dust storms occur during wet, moist and low-wind conditions. Please see here for examples:

Dupont, S., Klose, M., Irvine, M., González-Flórez, C., Alastuey, A. Bonnefond, J.-M., Dagsson-Waldhauserova, P., Gonzalez-Romero, A., Hussein, T., Lamaud, E., Meyer, H., Panta, A., Querol, X. Schepanski, S. Vergara Palacio, Wieser, A., Diez, J., Kandler, K., and Pérez García-Pando, C., 2024. Impact of dust source patchiness on the existence of a constant dust flux layer during aeolian erosion events. Journal of Geophysical Research: Atmospheres 129(12), e2023JD040657

Dagsson-Waldhauserova, P., Arnalds, O., Olafsson, H., Skrabalova, L., Sigurdardottir, G.M., Branis, M., Hladil, J., Skala, R., Navratil, T., Chadimova, L., von Lowis of Menar, S., Thorsteinsson, Th., Carlsen, H.K., and Jonsdottir I., 2014. Physical properties of suspended dust during moist and low wind conditions in Iceland. Icelandic Agricultural Sciences 27: 25 – 39.

Dagsson-Waldhauserova, P., Arnalds, O., Olafsson, H., Hladil, J., Skala, R., Navratil, T., Chadimova, L., Meinander, O., 2015. Snow-dust storm A case study from Iceland, March 7th 2013. Aeolian Research 16, 69–74.

**The introduction has been updated to reflect this.**

*Aeolian erosion is related to wind shear stress, which is related to the gradient of wind speed with height and the dynamical viscosity of the air (Marticorena, 2014). In some cases, dust emission may occur in low-wind conditions, as surface heating may cause sufficient upward air motion to lift and suspend silt-sized particles (Dagsson-Waldhauserova et al., 2014). In most cases however, a wind speed threshold must be met to overcome the forces holding particles in place on the surface, namely their weight, and interparticle cohesion forces resulting from electrostatics and soil moisture (Shao and Lu, 2000; Kok et al., 2012). These forces are especially strong for ultrafine soil particles on the order of microns or smaller. These particles are typically liberated by the impaction of larger particles under the forces of wind, a process called "saltation" (Kok et al., 2012).*

L152 – remove 'et al.'

**Corrected.**

L271 – Results in this chapter could be compared to results in Dupont et al. (2024)

Dupont et al. (2024). Impact of dust source patchiness on the existence of a constant dust flux layer during aeolian erosion events. Journal of Geophysical Research: Atmospheres 129(12), e2023JD040657

**We have now added some discussion of the Dupont et al. article in Section 3.2.**

*The mean diameter for the particle number distribution of 1.43 μm is comparable to, yet slightly smaller than, the 1.69 μm reported for an Icelandic mineral dust event where a similar method was employed (Dagsson-Waldhauserova et al., 2014). While Dupont et al. (2024) do not report comparative moments of the OPC-derived size distributions in their flux work, comparable features arise upon inspection as the number distributions also exhibit bi-modality, with a mode in the 0.3 – 0.6 μm range, and a second mode centered around 2 μm for measurements collected at 2.7 m from the ground (Dupont et al., 2024).*

L333 – RH seems to be quite low for the glacial and ice-proximal environment. Can you also explain the general RH conditions in this area? Is it normally 40-60% or was this exceptionally dry period?

**The region surrounding Lhù'ààn Mân' is described as a subarctic climate with cool summers and year-round precipitation, a Dfc climate zone in the Köppen-Geiger classification having little precipitation (Beck et al., 2018). This information has been added to the description of the sampling site (Section 2.1).**

*The region surrounding Lhù'ààn Mân' is described as a subarctic climate with cool summers and year-round precipitation, a Dfc climate zone in the Köppen-Geiger classification, having little precipitation (Beck et al., 2018).*

L468- Please change dot to comma

**Corrected.**

L472 – Add IPCC in the brackets of reference

**Corrected.**

L663 – Please correct the ending of the reference

**Corrected.**

**References:**

Kastury, F., Smith, E., Karna, R. R., Scheckel, K. G., and Juhasz, A.: An inhalation-ingestion bioaccessibility assay (IIBA) for the assessment of exposure to metal (loid) s in PM10, Science of the Total Environment, 631, 92-104, 2018.

Ranjbar, K., O'Neill, N. T., Ivanescu, L., King, J., and Hayes, P. L.: Remote sensing of a high-Arctic, local dust event over Lake Hazen (Ellesmere Island, Nunavut, Canada), Atmospheric Environment, 246, 10.1016/j.atmosenv.2020.118102, 2021.

Rencz, A. and Shilts, W.: Nickel in soils and vegetation of glaciated terrains, in: Nickel in the Environment, John Wiley & Sons, New York, 151-188, 1980.

Sayedain, S. A., O'Neill, N. T., Ranjbar, K., Gauvin-Bourdon, P., Chang, R., Hayes, P., and King, J.: Satellite-Based Remote Sensing of Local Dust Events Over the Canadian Arctic Archipelago, AGU Fall Meeting Abstracts, Washington, D.C., December 01, 20242024.